# Safety and Immunogenicity of the Nonavalent Human Papillomavirus Vaccine in Women Living with HIV

**DOI:** 10.3390/vaccines12080838

**Published:** 2024-07-25

**Authors:** Carmen Hidalgo-Tenorio, Raquel Moya, Mohamed Omar, Leopoldo Muñoz, Antonio SamPedro, Javier López-Hidalgo, Coral Garcia-Vallecillos, Patricia Gómez-Ronquillo

**Affiliations:** 1Infectious Disease Unit, Hospital Universitario Virgen de las Nieves, IBS-Granada, 18012 Granada, Spain; gvcoral@gmail.com (C.G.-V.); patriciagomez92@hotmail.com (P.G.-R.); 2Internal Medicine, Complejo Hospitalario de Jaén, 23007 Jaén, Spain; raquelmoyamegias@gmail.com; 3Infectious Disease Unit, Complejo Hospitalario de Jaén, 23007 Jaén, Spain; omarampa@gmail.com; 4Infectious Diseases Unit, Hospital Universitario San Cecilio, IBS-Granada, 18012 Granada, Spain; lmunnoz@yahoo.com; 5Microbiology Department, Hospital Universitario Virgen de las Nieves, IBS-Granada, 18012 Granada, Spain; antoniojsampedro@gmail.com; 6Pathology Department, Hospital Universitario Virgen de las Nieves, IBS-Granada, 18012 Granada, Spain; javierl.lopez.sspa@juntadeandalucia.es

**Keywords:** nonavalent HPV vaccine, high squamous intraepithelial lesion (HSIL), low squamous intraepithelial lesion (LSIL), WLHIV, anal cancer, cervical cancer

## Abstract

Background: The objectives were to evaluate the safety and immunogenicity of the nonavalent human papillomavirus (nHPV) vaccine in adult Spanish women living with HIV (WLHIV); the prevalence of anal and cervical dysplasia and nHPV vaccine genotypes in the anus and cervix; and risk factors for high-risk HPV (HR-HPV) infection in anal mucosa. Methods: In this single-center, open-arm, non-randomized clinical trial, the nHPV vaccine was administered at 0, 2, and 6 months to WLHIV enrolled between February 2020 and November 2023, measuring vaccine antibody titers pre-vaccination and at 2, 6, and 7 months after the first dose. Cervical and anal cytology and HPV PCR genotyping studies were performed. Women with abnormal cytology and/or anal or cervical HPV infection at baseline underwent high-resolution anoscopy and/or colposcopy. Results: A total of 122 participants were included with mean age of 49.6 years: 52.5% smoked; 10.7% had anal-genital condylomatosis; 38.5% were infected by HR-HPV in the anus and 25.4% in the cervix, most frequently HPV 16; 19.1% had anal intraepithelial neoplasia 1-(AIN1); and 3.1% had cervical intraepithelial neoplasia 1 and 2 (CIN1/CIN2). Vaccine administration did not modify viral–immunological status (CD4 [809 ± 226.8 cells/uL vs. 792.35 ± 349.95; *p* = 0.357]) or plasma HIV load (3.38 ± 4.41 vs. 1.62 ± 2.55 cop/uL [log]; *p* = 0.125). Anti-HPV antibodies ([IQR: 0–0] vs. 7.63 nm [IQR: 3.46–19.7]; *p* = 0.0001) and seroconversion rate (8.2% vs. 96.7% [*p* = 0.0001]) were increased at 7 versus 0 months. There were no severe vaccine-related adverse reactions; injection-site pain was reported by around half of the participants. HR-HPV infection in the anus was solely associated with a concomitant cervix infection (HR 5.027; 95% CI: 1.009–25.042). Conclusions: nHPV vaccine in adult WLHIV is immunogenic and safe.

## 1. Introduction

Anal squamous cell cancer (ASCC) is currently one of the most frequent AIDS-nondefining tumors in people living with HIV (PLHIV) [1], mainly in men who have sex with men (MSM) and in women, and its incidence is higher among women living with HIV (WLHIV) than among women in the general population [2]. ASCC and cervical cancer (CC) have been closely related to human papillomavirus (HPV) infection [3]. 

HPV is considered the most frequent sexually transmitted infection (STI), and it is estimated that more than 80% of sexually active persons will contract this infection at some point in their life [4]. One of the main factors related to mucosal HPV infection is the presence of HIV infection, and the associated disease burden is therefore greater than observed in the general population [5]. One study in North America found a significantly higher incidence of anal cancer (AC) in a cohort of PLHIV (North American AIDS Cohort Collaboration on Research and Design) than in the general population (60.1/100.000 vs. 1.2/100.000 person-years, respectively) [6]. Among PLHIV, the highest infection rate is observed in MSM [7], and the type of sexual relations and immunosuppressive status are known risk factors for high-risk HPV infection in PLHIV [8]. The mucosae most frequently infected with HIV in PLHIV are anal, followed by female genital and oropharyngeal mucosae [9].

CC was the most frequent AIDS-defining neoplasm in women living with HIV (WLHIV) before the implementation of highly active antiretroviral therapy (ART) and CC screening programs, which markedly reduced its incidence [10]. The impact of ART on AC is controversial, with an increased prevalence and incidence described by studies conducted in the early-ART period, possibly related to the longer survival of patients [11,12,13]; while more recent studies (late-ART period) in seropositive MSM described a certain protective effect of ART against high squamous intraepithelial lesions (HSILs) or precursor lesions and AC [14,15,16]. However, our investigation of HIV-infected women during the late-ART period (>2011) revealed an elevated prevalence and incidence of HSILs in the anus, higher than in the cervix, despite the greater clearance than acquisition of oncogenic genotypes in the anus [17]. No consensus has been reached by scientific societies on HSIL and AC screening protocols, including the selection of candidates for this procedure [18,19]. In Spain, the GESIDA consensus document on the diagnosis and treatment of STIs in adults, children, and adolescents recommends anal cytology studies in PLHIV with condylomas, MSM, and women with cervical dysplasia [18]. Nevertheless, although cytology is the most widely applied approach to HSIL and AC screening, its variable and limited sensitivity means that a non-negligible percentage of affected patients are not diagnosed [20,21,22].

Anal HSIL treatment has been recommended since publication of the ANCHOR study [23], but there is no agreement on the most appropriate strategy. Treatments range from local surgical excision (mucosectomy) to fulguration, radiofrequency, infrared, and topical treatments (e.g., imiquimod or cidofovir). ASCC is treated by local radiotherapy with or without chemotherapy and with or without radical surgery (abdominal–pelvic amputation), and it carries a poor prognosis [24]. Prevention measures are therefore crucial to avoid infection of the anal mucosa, including the utilization of condoms [25,26] and HPV vaccination [27]. 

The tetravalent HPV vaccine has demonstrated excellent safety and immunogenicity outcomes in adult PLWHIV [28,29]. However, a phase III clinical trial in PLHIV aged >36 years, including women and MSM, was interrupted because the vaccine did not prove effective to prevent HPV infection or the onset of premalignant anal mucosa lesions [30]. 

With this background, the working hypothesis of this study was that the nonavalent HPV vaccine is immunogenetic and safe in adult WLHIV residing in Spain. The objectives of this first-ever clinical trial of the nHPV vaccine (Gardasil-9 ^©^) in WLHIV were to assess its immunogenicity and safety, determine the prevalence of HPV infection and dysplasia in women’s genital and anal mucosae, and evaluate possible factors implicated in high-risk HPV (HR-HPV) infection of anal mucosa. 

## 2. Methods

### 2.1. Study Design

In this phase IV, multicenter, open-label, single-arm trial of nonavalent HPV (nHPV) vaccine, adult WLHIV were enrolled between 13 February 2020 and 6 November 2023. Participants had been referred to the infectious disease units of three Andalusian public hospitals (Virgen de las Nieves University Hospital of Granada, San Cecilio University Hospital of Granada, and Jaen City Hospital Complex). The principal researcher (C.H.T.) attended all study participants at the HPV office of Virgen de las Nieves University Hospital. The multidisciplinary research team comprised a pathologist (J.L.H.), a laboratory technician (M.G.), a microbiologist (A.S.), a trial coordinator (C.G.), internal medicine specialists (R.M. and P.G.), and infectious disease specialists (M.O., L.M., and C.H.T, who were responsible for data gathering, high-resolution anoscopy (HRA) studies, and group coordination). The trial coordinator (C.G.) oversaw the follow-up and scheduling of participants and the custody of their data. 

Inclusion criteria were WLHIV status, age ≥ 18 years, and the absence of simultaneous vaginal or anal infection by all seven HPV oncogenic genotypes in the nonavalent vaccine (HPV 16, 18, 31, 33, 45, 52, and 58). 

Exclusion criteria were an active opportunistic disease, the diagnosis of anal HSIL or AC at enrolment, and a history of aluminum or yeast allergy. 

Clinical trial registration: ID: NCT04270773.

https://classic.clinicaltrials.gov/ct2/show/NCT04270773 (accessed on 12 February 2020).

### 2.2. Intervention

Epidemiological, clinical, analytical, and pathological data were gathered in compliance with Spanish data protection legislation (Organic Law 3/5 December 2018). The study followed the principles of the Helsinki Declaration and was approved by the central biomedical research ethics committee of Andalusia (Decree 8/30 January 2020; Law 14/3 July 2007). This project, classified by AEMPS as a phase IV clinical trial post-authorization on 5 February 2020, was funded by the Carlos III Health Institute (#PI19/00283) and is registered at ClinicalTrials.gov ID: NCT04270773. https://classic.clinicaltrials.gov/ct2/show/NCT04270773. 

At the enrolment visit (V0), the study was described and explained before inviting participants to sign informed consent to their involvement and the performance of anal and cervical cytology (self-testing as instructed by the trial coordinator) and blood analyses, including anti-HPV antibodies, full blood count, biochemical values, lymphocyte count (CD4 and CD8), and HIV viral load. High-resolution anoscopy (HRA) was performed in patients with abnormal anal cytology and/or HPV infection. When the cervical self-testing procedure showed altered cytology and/or HPV infection, patients were referred to the gynecologist for repetition of the cytology, undergoing a colposcopy if self-testing results were confirmed. 

Patients who met inclusion criteria and had signed informed consent received the first vaccine dose at the baseline visit (V1), the second dose at two months (V2, 8 weeks after the first), and the third dose at six months (V3, 24 weeks after the first). At V1, V2, V3, and V4 (28 weeks after the first dose), a blood sample was drawn before the dose administration to obtain the same analytical data as at V0, and patients completed a questionnaire on adverse effects (AEs) at 15 min after the vaccination. 

### 2.3. Clinical–Epidemiological Variables 

At V1, data were gathered on age, number of different sexual partners during the previous 12 months and during their lifetime; practice or not of anal sex; age at onset of sexual life; utilization of condoms (qualitative and percentage); employment status (active or retired); schooling; smoking habit (packs/year); alcohol consumption (in standard units of alcohol (SUA); intravenous drug addiction; months since HIV diagnosis; CDC stage of HIV; CD4 nadir; months under ART; current ART line; virological failure; concomitant treatment; presence of other diseases, including STIs, syphilis, cytomegalovirus (CMV), IgG, or condylomas; chronic infection by hepatitis B virus (HBV) or hepatitis C virus (HCV); history of anal and/or genital condylomatosis and its treatment; and history of cervical or anal disease. 

At V2, V3, and V4 (2, 6, and 7 months), information was collected on number of different partners; practice of anal sex; utilization of condoms (qualitative and percentage); ART consumption (adherence, switch, and failure); onset of STIs or condylomas; and questionnaire responses on vaccine-related AEs (local reaction, fever, nausea, vomiting, dizziness, syncope, headaches, allergic reactions, pruritus, breathing difficulties/sounds, urticaria lymphadenopathies, chest/lower limb pain, confusions, shivers, and muscle pain), which were classified into four grades. Participants with grade 4 AE were referred to the pharmacovigilance Yellow Card system. 

### 2.4. Analytical Variables 

At V1, V2, V3, and V4, data were obtained on full blood count, biochemical values, CD4 and CD8 lymphocyte counts, HIV viral load, anti-HPV antibodies, syphilis, HCV, HBV, CMV IgG, and VEB IgG serology. 

### 2.5. PCR Study of HPV and Anal and Cervical Cytology

At enrolment (V0), two samples of anal and cervical mucosae were self-collected, using saline-soaked cotton swabs that were immersed in liquid medium (thin-layer liquid), performing the Linear Array HPV Genotyping Test in a thermocycler (GeneAmp PCR System 9700, Applied Biosystems, Roche, Basel, Switzerland) for HPV detection and genotyping by in vitro qualitative polymerase chain reaction (PCR) and using the Thin Prep 2000 (Hologic) Processor for the cytology study. Both samples were sent to the hospital pathology laboratory, where a single pathologist (J.L.H.) performed the cytology study and validated the HPV PCR results. Genotypes 16, 18, 26, 31, 33, 35, 39, 45, 51–53, 56, 58, 59, 66, 68, 73, and 82 were considered high-risk (HR-HPV). Genotypes 6, 11, 34, 40, 42–44, 54, 55, 57, 61, 70–72, 81, 83, 84, and 89 were considered low-risk (LR-HPV). Viruses were also classified as HPV 18 (39, 45, 59, and 68) or HPV 16 (31, 33, 35, 52, 58, and 67) species [31]. 

The cytology used the Bethesda classification of four types of lesions [32]: atypical squamous cell (ASC), ASC that cannot rule out high-grade lesion (ASC-H), low squamous intraepithelial lesion (LSIL), and high squamous intraepithelial lesion (HSIL). The LAST 2014 histological classification was used to categorize anal lesions as LSIL (anal intraepithelial neoplasia [AIN1]/condyloma), HSIL (AIN2, AIN3), or anal canal carcinoma (ACC) [33]. 

### 2.6. High-Resolution Anoscopy

At enrollment (V0), WLHIV with abnormal anal cytology and/or HPV infection underwent HRA with biopsy after instillation of acetic acid and Lugol’s iodine, using a Zeis 150 FC colposcope (c) to take samples from aceto-Lugol-negative zones and recording the specimen with the most severe dysplasia. 

### 2.7. Gynecological Examination 

Participants with positive HPV PCR and/or abnormal self-testing cervical cytology were referred to a gynecologist for examination, cytology, and colposcopy to verify self-testing results. 

### 2.8. Nonavalent HPV Vaccine

The nonavalent vaccine (Gardasil-9©) (Merck Research Laboratories, Darmstadt, Germany) was administered intramuscularly in deltoid muscle at V1, V2, and V3 by the principal researcher (C.H.T.) in the Infectious Diseases office. 

### 2.9. Antibodies against the 9 HPV Genotypes of the Vaccine 

Anti-HPV antibodies were analyzed at V0, V2, V3, and V4 by a single specialist (A.S.) in the Microbiology Department of Virgen de las Nieves Hospital. The HPV IgG ELISA EIA-4907 kit (DIA.PRO, Milan, Italy, Weldon Biotech) was used to detect antibodies against the main protein of HPV capsid (L1) in accordance with the manufacturer’s instructions. Briefly, the solid phase was treated with diluted samples in the first incubation, when any anti-HPV IgG were captured by the antigens. After washing out all other components of the sample, bound anti-HPV IgG were detected in the second incubation by adding peroxidase (HRP)-labeled anti-IgG antibodies. Action of the enzyme captured in the solid phase on the substrate/chromogen mixture generated an optical signal proportional to the amount of anti-HPV IgG antibodies in the sample. A cut-off value was applied to classify the optical density as positive or negative. Blood samples were centrifuged and frozen at −20 °C until their analysis, performed using an ELISA for semi-quantitative determination of IgG class antibodies against HPV in human plasma and sera, which is only available for research purposes. Results were recorded as positive or negative and expressed (in nm) as median values with inter-quartile range (IQR).

### 2.10. Statistical Analysis 

#### 2.10.1. Sample Size

The sample size was estimated based on our previous observation of anal infection by HR-HPV in 49.4% of this study population. A sample size of 123 WLHIV was calculated for a statistical power of 80% and significance level of 5% to reject the null hypothesis that the post-vaccination prevalence would not exceed 51.40%, with a non-superiority limit of 7%. 

#### 2.10.2. Data Analysis 

In a descriptive analysis, quantitative variables were expressed as mean values with standard deviation (SD) and median values with IQR and qualitative variables as absolute and relative frequencies. Prevalences of HPV and anal and cervical mucosae dysplastic lesions were calculated with 95% confidence interval. 

Bivariate analyses were conducted to study variables related to anal mucosa infection by HR-HPV genotypes, applying the Kolmogorov–Smirnov test to check the normality of their distribution. The Student’s *t*-test was used for independent quantitative variables with normal distribution, the Mann–Whitney U test for those with non-normal distribution, and the Wilcoxon test for related quantitative variables. Pearson’s or Fisher’s chi-square test was used for qualitative variables as appropriate. Logistic regression multivariate analysis was conducted, entering variables showing statistical significance in bivariate analysis and other factors considered clinically relevant (age, sexual relations in previous 12 months, condom utilization, latent CMV infection, CD4 nadir, undetectable viral load, CD4 count at enrolment, history of AIDS, ART duration, previous ART lines, and history of cervical dysplasia). Variables were entered using a forward stepwise procedure, with a *p*-value of 0.05–0.10 for each entry. *p* < 0.05 was considered significant in all tests. SPSS version 20.0 (IBM SPSS, Armonk, NY, USA) was used for data analyses.

## 3. Results 

### 3.1. Study Population

Out of the total screened population of 139 WLHIV, study eligibility criteria were met by 122, who were enrolled in the study (Figure 1). Their mean (SD) age was 49.6 (9.51) years, and 71.3% held Spanish nationality. The median number of sexual partners over the previous 12 months was one (IQR: 0–1); the median number of sexual partners since starting sexual relations was five (IQR: 3–12); 43.4% used condoms; 52.5% were active smokers; 1.6% had active chronic infection by HBV and 0.8% by HCV; 4.9% had syphilis; 7.3% other STIs; and 10.7% had anal–genital condylomatosis. Other epidemiological data are reported in Table 1.

### 3.2. HPV PCR and Anal Cytology

Among the 122 women included, 42 had anal infection (34.4%) by low-risk genotypes, 47 (38.5%) by high-risk genotypes, and 30 (24.6%) mixed infection. The most prevalent genotypes in anal mucosa were HPV 16 (10.7%) and HPV 51 (8.2%), followed by HPV 6, 62/81, and 68 (7.4% for each). Among genotypes covered by the nonavalent vaccine, the highest prevalence in anal mucosa was for genotype 16 (10.7%), followed by low-risk HPV 6 (7.4%), HPV 45, 52, and 58 (4.9% for each), HPV 33 (3.3%), HPV18 (1.6%), and finally HPV 31 and 11 (0.8% for each) (Table 2). Appendix A exhibits the VPH genotypes isolated in anal mucosa that are not included in the Gardasil-9 vaccine.

The anal cytology results show that 80 (65.6%) were normal, 21(17.2%) had LSIL, and 19 (15.6%) had ASCUS. The HRA results in 96 participants found that 76 (80.8%) were normal and 18 (19.1%) had LSIL (AIN1). The remaining results are exhibited in Table 2. 

### 3.3. Cytology and Cervical HPV PCR 

Among the 122 participants, low-risk genotypes were detected in cervical mucosa samples from 31 (25.4%), high-risk genotypes in those from 32 (26.2%), and mixed infection by high- and low-risk viruses in those from 13 (10.7%). The most prevalent genotypes were HPV 16 and 62/81 (7.4% for each) and 6 and 42 (6.6% for each). Among the genotypes covered by the nonavalent vaccine, the highest prevalence was for genotype 16 (7.4%), followed by HPV 6 (6.6%), HPV 31, 33, and 58 (1.6% for each), and genotypes HPV 45 and 52 (0.8% for each). No infection by genotypes 18 or 11 was detected in any sample. Appendix A displays the HPV genotypes isolated in cervical mucosa that are not included in the Gardasil-9 vaccine.

Cervical cytology was normal in 111 (91%), while 4 (3.3%) had LSIL, 1 (0.8%) HSIL, and none (0%) ASCUS. The gynecology study in 64 (52.5%) of the women showed that 60 (93.8%) were normal, while 2 (3.1%) had CIN1 and 2 (3.1%) CIN2. The remaining results are summarized in Table 3. 

In comparison to the cervical mucosa samples, anal mucosa samples had higher prevalences of high-risk HPV genotypes (38.5 vs. 26.2%; *p* = 0.005), low-risk genotypes (34.4 vs. 25.4%; *p* = 0.009), and coinfection by high- and low-risk genotypes (24.6 vs. 10.7%; *p* = 0.0001).

### 3.4. Adverse Effects

The vaccination schedule was completed by 120 of the 122 women. Two women did not receive the third vaccine but provided blood samples for analysis at V4.

The AE rate significantly declined over the treatment period (*p* = 0.05). AEs were reported by 81 (66.4%) women after the first vaccine dose, most frequently injection-site pain (in 54.1%) with a median VAS score of 1 (IQR: 0–5). AEs were reported by 64 (52.4%) women after the second dose, most frequently injection-site pain 54 (44.3%), with a median VAS score of 0 (IQR: 0–3), and AEs were reported by 58 (48.3%) women after the third dose, again mainly injection-site pain 47 (38.5%), with a median VAS score of 0 (IQR: 0–3). A highly significant reduction in pain was observed between the first and third doses (54.1% vs. 38.5%, *p* = 0.008). The remaining results are displayed in Table 4. No participant had a vaccine-related AE that was classified as severe (Grade 3 or 4) or caused them to miss the next dose.

The vaccine treatment produced no changes in CD4 count (809.34 ± 367.36 cells/uL vs. 792.35 ± 349.95; *p* = 0.357) or plasma HIV viral load (3.38 ± 4.41 vs. 1.62 ± 2.55 cop/uL [log]; *p* = 0.125) between baseline and seven months. No other analytical results were outside the range of normality (Table 5 and Figure 2a). 

### 3.5. Immunogenicity

The vaccine treatment produced a significant increase (*p* = 0.0001) in total IgG percentage against HPV, finding anti-HPV antibodies in 8.2% of the women at baseline, in 59% at 2 months after the first dose, in 89.3% at 6 months, and in 96.7% at 7 months. It also produced a significant increase (*p* = 0.0001) in IgG levels (0 nm [IQR: 0–0.14] at baseline vs. 6.91 nm [IQR: 3.3–19.7] at 7 months; (Table 5 and Figure 2b). 

### 3.6. Risk Factors of Anal Infection by HR-HPV

In bivariate analyses, anal infection due to high-risk genotypes was related to a larger number of sexual partners (6 [IQR: 4–14] vs. 3 [IQR: 2–9], *p* = 0.04); shorter time since HIV diagnosis (12.5 years [IQR: 5.4–201] vs. 22.7 years [14.3–27]; *p* = 0.03); infection by low-risk genotypes in anal mucosa (62.5 vs. 18.1%; *p* = 0.0001) and larger number of these genotypes (1 [IQR: 0–1] vs. 0 [0–0], *p* = 0.0001); infection by high-risk genotypes in genital mucosa (43.8 vs. 13.7%, *p* = 0.0001) and larger number of these genotypes (1 [0–1] vs. 0 [0–0], *p* = 0.0001); and infection by LR-HPV genotypes (39.6 vs. 16.4%, *p* = 0.004) and larger number of these genotypes (1 [IQR: 0–1] vs. 0 [0–0], *p* = 0.02). The sole risk factor to emerge from the multivariate analysis was a concomitant cervical infection by high-risk HPV genotypes (hazard ratio = 5.027; 95% CI, 1.009–25.042) (Appendix A).

## 4. Discussion

The WLHIV in this study had a mean age of around 50 years. They typically had Spanish nationality and had lived with a single stable partner over the previous year, and more than half of them were smokers. Around one-fifth had a history of gynecological and anal dysplasia, and a few had a history of other STIs (e.g., syphilis, gonorrhea, or chlamydia). They had been diagnosed with HIV around two decades earlier and had received ART for a mean of 15 years, so that their viral–immunological status was excellent, with only one woman in virological failure. Among the women with an HPV infection of the anal mucosa, more than one-third were infected by high- and low-risk HPV genotypes, and around one-fifth had AIN1; target genotypes of the nonavalent vaccine were detected in less than 5%, except for genotypes 6 and 16 (7.4 and 10.7%, respectively). Among those with a cervical mucosa infection, the most prevalent target genotypes were again HPV 6 and HPV 16, but their prevalence was lower than in the anal mucosa. The majority of colposcopies were normal, with only 3.1% having CIN 1 and 3.1% CIN2. A previous study in Spanish women found that the fraction of high-grade noninvasive cervical intraepithelial squamous lesions attributable to the target genotypes of the nonavalent vaccine was 86% in those with CIN2 and 86% in those with CIN3/carcinoma in situ in all age groups, although mainly in over-35-year-olds [34].

The HPV vaccine is considered most advantageous when administered before the initiation of sexual relations and up to the age of 26 years (between 12 and 26 years). Its effectiveness in women has been positively associated with younger age at vaccination. Thus, an observational cohort study of vaccinated women in England found that the risk of developing CIN3 was reduced by 39% (95% CI 36–41) when their age was between 16 and 18 years, 75% (72–77) when between 14 and 16 years, and 97% when between 12 and 13 years [35]. Nevertheless, the SPERANZA study observed that the tetravalent HPV vaccine was also beneficial in adult sexually active women with CIN2+ after cervical conization, achieving reductions in CIN2+ relapse rate (1.2 vs. 6.4%) and relapse rate risk (81.2% [34.3–95.7]). The prospective case–control study detected HPV genotypes 11, 16 (63.6%), 18, 31, 33, 45, 53, and 82 in the non-vaccinated women who relapsed but only HPV genotypes 33 and 82 in the vaccinated women, who had none of the target genotypes of the tetravalent vaccine, indicating an effectiveness of 100%. The authors concluded that sustained clinical effectiveness against the relapse of high-grade cervical lesions was obtained by vaccination after cervical conization as adjuvant to surgical treatment [36].

In the present WLHIV, the nonavalent vaccination schedule caused no relevant clinical changes in the CD4 count, HIV viral load, or key analytical parameters (full blood count and biochemistry), and it produced significant increases in the percentage of total IgG against HPV and IgG levels. More than half of the women reported an AE after the first dose, most frequently pain at the injection site, but the proportion with an AE and their pain levels were significantly lower after the third dose, and no AE was severe (grade 3 or 4). In a clinical trial of the tetravalent HPV vaccine in Spain, our group also found no relevant vaccination-related viral–immunological, clinical, or analytical alterations in adult MSM-LHIV, who showed a significant increase in the level of antibodies against target HPV genotypes [37]. Two meta-analyses on the immunogenicity, safety, and effectiveness of bivalent, quadrivalent, and nonavalent HPV vaccines reported a solid and safe immune response and a reduction over time in antibody titers and seropositivity rates, although these continued to be elevated [38,39]. The seroconversion rate was around 100% for each vaccine, and the level of antibodies against target HPV genotypes was significantly higher in the vaccinated versus placebo groups, which did not differ in severe AE rate, T CD4 + cell count, or HIV viral load [34]. The results of these meta-analyses support the HPV vaccination of PLHIV and highlight the need for further clinical trials to determine the efficacy of the HPV vaccine against infections and neoplasms. Given their burden of HPV disease, the protection against HPV 18 may be less robust in PLHIV, but the consistent immune response suggests that they may benefit from HPV vaccination after acquiring HIV [38,39]

The sole risk factor for anal infection by oncogenic HPV genotypes in the participants was concomitant infection by these genotypes in the cervix. The rate of infection by HR-HPV genotypes was significantly higher in anal versus cervical mucosae, possibly because their clearance rate is higher than their acquisition rate in the cervix [17], and the anus acts as a reservoir for cervical HPV. HPV is the most frequent STI worldwide and will infect most people during their lifetime; however, it is expected to disappear in the general population, but not in PLHIV. HPV infection is known to be a causal agent of cervical cancer, and there is also increasing evidence that the infection and/or related cervical inflammation increases the risk of HIV infection, implicating the two viral infections in mutual spread [40]. In addition, HIV would play a role in the higher incidence and greater persistence of HPV infection in the anal canal and cervix of PLHIV in comparison to the general population.

In conclusion, the nHPV vaccine is safe and immunogenic in adult WLHIV. The sole risk factor for anal mucosa infection due to high-risk genotypes in this population was concomitant cervical infection by the same genotypes.

The limitations of this trial derive from its design (single-arm, single-center study in older WLHIV), preventing extrapolation of the data to other populations. However, it is the first clinical trial published to date on the nHPV vaccine in WLHIV and corroborates the safety and immunogenicity of HPV vaccination in this specific population.

## Figures and Tables

**Figure 1 vaccines-12-00838-f001:**
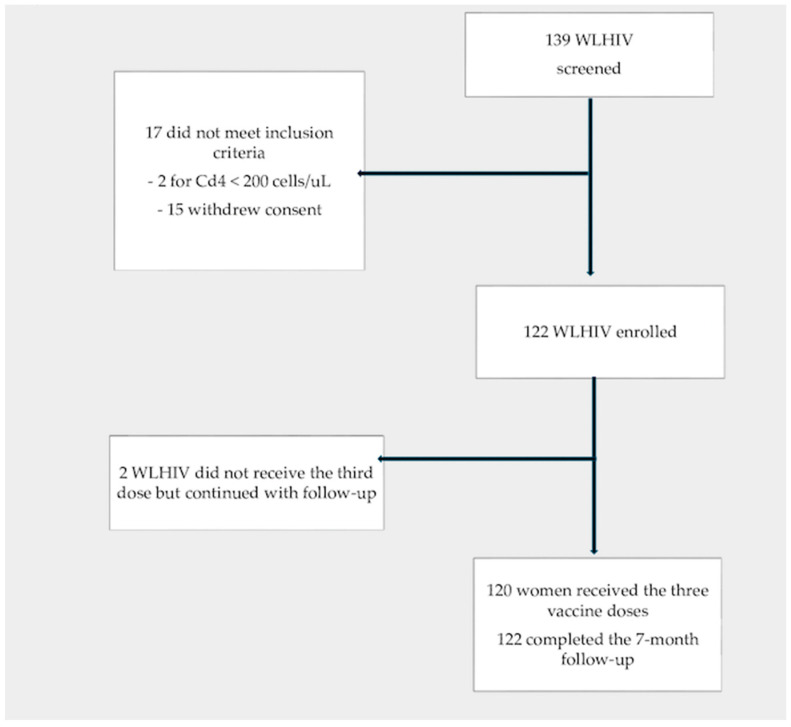
Flowchart of the clinical trial.

**Figure 2 vaccines-12-00838-f002:**
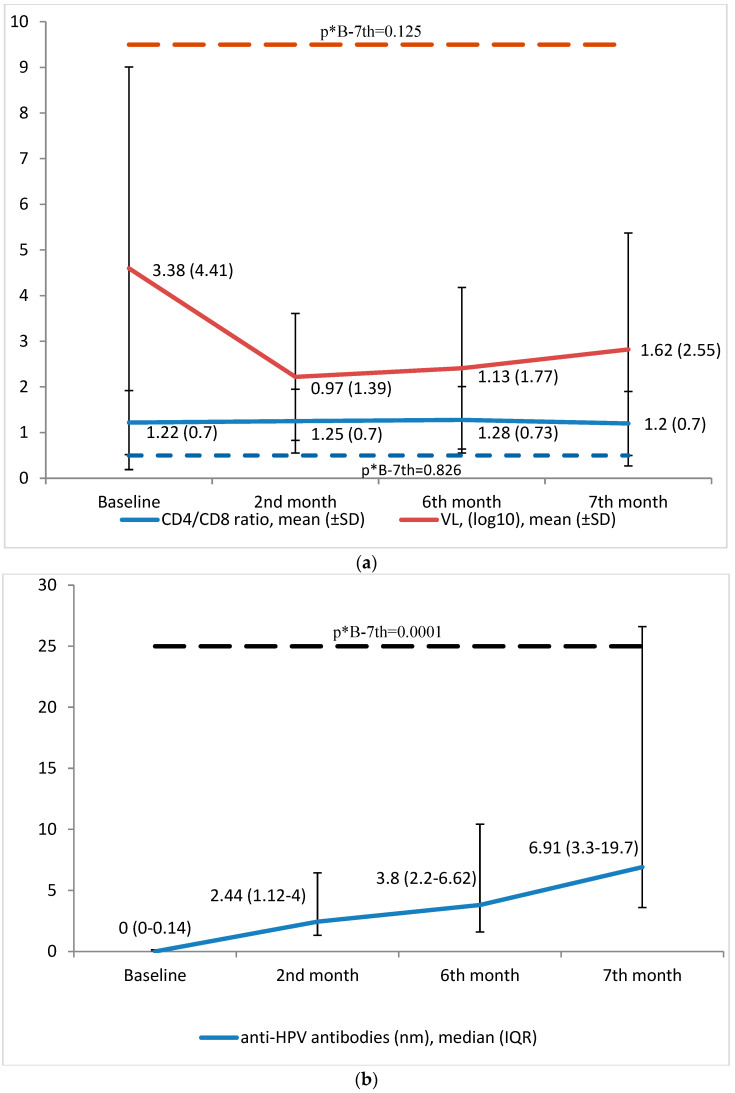
(**a**). CD4/CD8 ratio and HIV viral load levels over 7 months of follow-up. (**b**). CD4/CD8 ratio and HIV viral load levels over 7 months of follow-up. *p** B-7th = significance between baseline and 7th month.

**Table 1 vaccines-12-00838-t001:** Baseline demographics of WLHIV included in clinical trial.

Variables	n = 122
Age, years; mean (±SD)	49.62 (±9.51)
Spanish nationality, n (%)	87 (71.3)
University education, n (%)	10 (8.2)
Partners in the previous 12 months, median (IQR)	1 (0–1)
Life-time partners number, median (IQR)	5 (3–12)
Months of sexual activity, median (IQR)	390 (288–456)
History of sex worker, n (%)	20 (16.4)
Sex worker currently, n (%)	2 (1.6)
Condom use, n (%)	5 (4.1)
History of anal/genital condylomas, n (%)	13 (10.7)
History of gynaecologic dysplasia	28 (22.9)
CIN1/VIN1/VAIN1	9 (7.4)
CIN2/CIN3/VIN2/3/VAIN2/3	15 (12.2)
Cervical/vaginal/vulvar cancer	2 (1.6)
History of anal dysplasia	33 (27)
AIN1	24 (19.7)
AIN2/3	7 (5.7)
Anal cancer	2 (1.6)
Duration of HIV, mean years (IQR)	19 (8–24)
History of AIDS, n (%)	58 (47.5)
CD4 mean nadir, cells/µL, (±SD)	226.9 (178.3)
CD4 mean, cells/µL (±SD)	809.3 (367.4)
CD8 mean, cells/µL (±SD)	794.3 (458.2)
CD8/CD4 ratio, mean (±SD)	1.2 (0.7)
VL of HIV log10, copies/mL (±SD)	3.38 (4.41)
VL < 50 copies/mL, n (%)	115 (94.3)
Virological failure, n (%)	3 (2.5)
Median duration of ART, years (IQR)	15 (8–21.2)
Number of lines of ART, median (IQR)	4 (3–6)
Antibodies of HPV, n (%)	7 (5.7)
Syphilis treated, n (%)	6 (4.9)
Other STD, n (%)	9 (7.3)
Chronic HCV infection not treated, n (%)	1 (0.8)
Chronic active HCV infection (AgHBs), n (%)	2 (1.6)
Smoking, n (%)	66 (55.1)
Smoking, packets/year, median (IQR)	10 (1.5–23)
Ex-IVDA, n (%)	27 (22.1)

Notes to Table 1. HCV: hepatitis C virus; HPV; human papilloma virus; Ex-IVDA; ex-intravenous drug abuser; SD: standard deviation; IQR: interquartile range.

**Table 2 vaccines-12-00838-t002:** Baseline anal HPV PCR, cytology, and HRA* results of participants enrolled.

Variables	n = 122
PCR of HPV, n (%)	
LR-HPV, n (%)	42 (34.4)
HR-HPV, n (%)	47 (38.5)
HR and LR HPV, n (%)	30 (24.6)
Number of HR-HPV (IQR)	0 (0–1)
Number of LR-HPV (IQR)	0 (0–1)
HPV Genotypes, n (%)	
Vaccine HPV genotypes	
HPV 6	9 (7.4)
HPV11	1 (0.8)
HPV16	13 (10.7)
HPV18	2 (1.6)
HPV 31	1 (0.8)
HPV 33	4 (3.3)
HPV 45	6 (4.9)
HPV 52	6 (4.9)
HPV 58	6 (4.9)
Cytology, n (%)	122 (100)
Normal	80 (65.6)
LSIL	21 (17.2)
HSIL	0 (0)
ASCUS	19 (15.6)
ASCUS-H	0
Invalid results	2 (1.4)
HRA, n (%)	96 (78.7)
Normal	76 (80.8)
LSIL (AIN1)	18 (19.1)

Notes to Table 2: LSIL: low squamous intraepithelial lesion; HSIL: high squamous intraepithelial lesion; ASCUS, atypical squamous cells undetermined significance, HPV: human papillomavirus; HR-HPV: high-risk HPV; LR-HPV: low-risk HPV; HRA*: high-resolution anoscopy.

**Table 3 vaccines-12-00838-t003:** Baseline cervical HPV PCR, cytology, and colposcopy results of patients enrolled.

Variables	n = 122
PCR of HPV	
LR-HPV, n (%)	31 (25.4)
HR-HPV, n (%)	32 (26.2)
HR and LR HPV, n (%)	13 (10.7)
Number of HR-HPV (IQR)	0 (0–1)
Number of LR-HPV (IQR)	0 (0–1)
HPV Genotypes, n (%)	
Vaccine HPV genotypes	
HPV6	8 (6.6)
HPV11	0 (0)
HPV16	9 (7.4)
HPV18	0 (0)
HPV 31	2 (1.6)
HPV 33	2 (1.6)
HPV 45	1 (0.8)
HPV 52	1 (0.8)
HPV 58	2 (1.6)
Cytology, n (%)	122 (100)
Normal	111 (91)
LSIL	4 (3.3)
HSIL	1 (0.8)
ASC/ASC-H	0 (0)
Invalid result	1 (0.8)
Colposcopy, n (%)	64 (52.5)
Normal	60 (93.8)
CIN1	2 (3.1)
CIN2	2 (3.1)
CIN3	0 (0)
Cervical cancer	0 (0)

**Table 4 vaccines-12-00838-t004:** On-treatment safety and tolerability.

Adverse Events	V1 n = 122	V2 n = 122	*p** (V1–V2)	V3 n = 120	*p** (V1–V3)
Total AE, n (%)	81 (66.4)	64 (52.4)	0.07	58 (48.3)	0.05
Injection-site pain VAS (0–10), median (IQR)	66 (54.1) 1 (0–4.5)	54 (44.3) 0 (0–3)	0.109	47 (38.5) 0 (0–3)	0.008
Local itching	6 (4.9)	4 (3.3)	0.066	3 (2.5)	0.102
Injection-site nodule	7 (5.7)	5 (4.1)	0.56	5 (4.1)	0.782
Dizziness	2 (1.6)	1 (0.8)	0.32	3 (2.5)	0.157
AE leading to treatment discontinuation, n (%)	0 (0)	0 (0)		0 (0)	
Deaths, n (%)	0 (0)	0 (0)		0 (0)	
Serious AE, n (%)	0 (0)	0 (0)		0 (0)	
Grade 3 or 4 abnormalities related to nHPV vaccine, n (%)	0 (0)	0 (0)		0 (0)	

*p**: significance and *p* < 0.05; VAS: visual analog scale.

**Table 5 vaccines-12-00838-t005:** Analytical results during the 7 months.

Variables	Baseline (B)	2 m	6 m	7 m	*p** B-7 m
CD4, cells/ul, mean (±SD)	809.34 (367.36)	834.63 (366.78)	812.08 (368.5)	792.35 (349.95)	0.357
CD8, cells/ul, mean (±SD)	790.59 (460.44)	817.05 (527.74)	815.78 (663.69)	809.61 (521.71)	0.534
CD4/CD8 ratio, mean (±SD)	1.2 (0.7)	1.25 (0.7)	1.28 (0.73)	1.2 (0.7)	0.826
VL, log10, mean (±SD)	3.38 (4.41)	0.97(1.39)	1.13(1.77)	1.62(2.55)	0.125
SGOT, mean (±SD)	21.24 (7.82)	22.28 (8.51)	22.39 (9.98)	23.63 (16.85)	0.051
SGPT, mean (±SD)	20.76 (10.41)	20.84 (8.69)	22.31 (11.20)	23.84 (21.23)	0.012
GGT, mean (±SD)	39.56 (120.12)	37.34 (110.53)	48.13 (218.51)	49.62 (189.40)	0.060
AP, mean (±SD)	77.85 (47.03)	79.06 (58.88)	79.15 (52.32)	83.45 (56.18)	0.191
Bilirubin, mean (±SD)	0.71 (1.01)	0.61 (0.62)	0.56 (0.27)	0.57 (0.25)	0.036
Creatinine, mean (±SD)	0.81 (0.19)	0.84 (0.48)	0.84 (0.51)	1.49 (±6.81)	0.64
Anti-HPV Ab, nm, median (IQR)	0 (0–0.14)	2.44 (1.12–4)	3.8 (2.2–6.62)	6.91 (3.3–19.7)	0.0001
Anti-HPV Ab positive, n (%)	10 (8.2)	72 (59)	109 (89.3)	118 (96.7)	0.0001

Notes to Table 5. Treatment-emergent grade 3 or 4 abnormalities defined by laboratory values: ALT > 5.0 x upper limit of normal (ULN); AST > 5.0 x ULN; Total bilirubin >2.5 x ULN; VL: viral load of HIV; SGOT: serum glutamic oxaloacetic transaminase; SGPT: serum glutamate pyruvate transaminase; GGT: gamma-glutamyltransferase; AP: alkaline phosphatase. Anti-HPV ab: anti-HPV antibodies. *p**: significance and *p* < 0.05.

## Data Availability

ClinicalTrials.gov ID: NCT04270773.

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
