# Peer review of "Safety and Immunogenicity of the Nonavalent Human Papillomavirus Vaccine in Women Living with HIV"

_vaccines, 2024, doi:10.3390/vaccines12080838_

Round 1

Reviewer 1 Report

Comments and Suggestions for Authors

In this study, Hidalgo-Tenorio et al have demonstrated safety and immunogenic profile of nonavalent human papilloma virus vaccine in patients with HIV. This study is very well done and is part of large multicenter trial cohort. This is very interesting study to delineate effects of this vaccination and its effect on patient health. I have some minor suggestions

-The data is mainly expressed in form of tables which will hardly attract the readers. I can propose to show only significant data in main text in small tables and preferably in figures and put some tables in supplemental.

-The authors did look into CD4 and CD8 T cells where no change was observed, however, authors should mention about B cells and plasma cells during vaccination.

 - To make the article interesting for general public, I would propose to modify the introduction with more details of papillomavirus infections in context of HIV and broad.

Comments on the Quality of English Language

The quality of english language is good with some minor corrections which will be covered with typoediting.

Author Response

Reviewer 1

Comments and Suggestions for Authors

In this study, Hidalgo-Tenorio et al have demonstrated safety and immunogenic profile of nonavalent human papilloma virus vaccine in patients with HIV. This study is very well done and is part of large multicenter trial cohort. This is very interesting study to delineate effects of this vaccination and its effect on patient health. I have some minor suggestions.

-The data is mainly expressed in form of tables which will hardly attract the readers. I can propose to show only significant data in main text in small tables and preferably in figures and put some tables in supplemental.

Response:  As recommended, we have reduced the size of Tables 2 and 3 by transferring data to supplementary material (S1 and S2) and have moved table 6 to supplementary material (S3). We also include a new figure depicting the time course of viral load, CD/CD8, and HPV antibody values over the study period.   

-The authors did look into CD4 and CD8 T cells where no change was observed, however, authors should mention about B cells and plasma cells during vaccination.

Response: This is a good suggestion; however, as can be seen in Material and Methods, we did not measure B cells or specific plasma cells in our conventional blood count.

 - To make the article interesting for general public, I would propose to modify the introduction with more details of papillomavirus infections in context of HIV and broad.

Response: We now provide more details in the following new paragraph in the Introduction:

HPV is considered the most frequent STI, and it is estimated that more than 80% of sexually active persons will contract this infection at some point in their life (4).  One of the main factors related to mucosal HPV infection is the presence of HIV infection, and the associated disease burden is therefore greater than observed in the general population (5). One study in North America found a significantly higher incidence of anal cancer in a cohort of PLHIV (North American AIDS Cohort Collaboration on Research and Design) than in the general population (60.1/100.000 vs 1.2/1000 person-year, respectively) (6). Among PLHIV, the highest infection rate is observed in MSM (7), and the type of sexual relations and immunosuppressive status are known risk factors for high-risk HPV infection in PLHIV (8). The mucosae most frequently infected with HIV in PLHIV are anal, followed by female genital and oropharyngeal mucosae (9)

Comments on the Quality of English Language

The quality of English language is good with some minor corrections which will be covered with typoediting.

Response: All of these errors have been corrected.

Reviewer 2 Report

Comments and Suggestions for Authors

The study evaluated the safety and immunogenicity of human papillomavirus nonavalent vaccine in HIV-infected women.  It is reported that people living with HIV are at higher risk of acquiring HPV as compared to healthy people. Hence, evaluation of such vaccine in the HIV-infected candidates is vital. The authors presented various data on the safety and immunogenicity of the HPV nonavalent vaccine. However, the way of the presentation of the results must be improved. The following are the suggestions or corrections that need to be made.

1. The authors need to expand the acronyms when mentioned for the first time. For example, AIN1, CIN1, and CIN2 in the abstract.

2. There is a missing "of" in-between "out" and "the total" in line 217.

3. Correct the spelling of "cells" in cells/uL in Figure 1. 

4. What are the major adverse effects observed after vaccination? The authors must mention those effects clearly in the adverse effects section 3.4.

5. V4 mentioned in line 266 should be corrected to V3.

6. Besides mentioning the percentage of individuals having antibodies against the HPV after the vaccination as mentioned in the Immunogenicity section 3.5, the presentation of antibody titers at different sample collection time points in the form of the figure should be included. The authors have mentioned the same in the table 5. However, the depiction of such important data in the form of a figure/graph would be better. The manuscript already has a lot of tables. The inclusion of some figures especially important data would improve the quality of the presentation and hence interest to the readers.

7. The authors should discuss the significant changes in the clinical variables like SGPT and Bilirubin (Table 5) after the vaccination. This could indicate some adverse effects on the liver functions. The authors have not mentioned these significant changes anywhere in the manuscript. This is very necessary considering the safety of the vaccine.

8. The data like CD4/CD8 ratio and viral load seem to have strong fold changes after the vaccination but no significant. This could be due to higher variation between the individuals. I would suggest presenting those data in graphical form like a scatter plot to make the reader easily understandable. The way of presentation of results can be improved by expressing them in figures whichever is possible instead of many tables. Figures reflect the data better than the tables and make it more easily understandable.

Author Response

The study evaluated the safety and immunogenicity of human papillomavirus nonavalent vaccine in HIV-infected women.  It is reported that people living with HIV are at higher risk of acquiring HPV as compared to healthy people. Hence, evaluation of such vaccine in the HIV-infected candidates is vital. The authors presented various data on the safety and immunogenicity of the HPV nonavalent vaccine. However, the way of the presentation of the results must be improved. The following are the suggestions or corrections that need to be made.

  1. The authors need to expand the acronyms when mentioned for the first time. For example, AIN1, CIN1, and CIN2 in the abstract.

Response: This has been done throughout the study.

  1. There is a missing "of" in-between "out" and "the total" in line 217.

Response: Corrected

  1. Correct the spelling of "cells" in cells/uL in Figure 1.

 Response: Corrected.

  1. What are the major adverse effects observed after vaccination? The authors must mention those effects clearly in the adverse effects section 3.4.
  2. Response: All adverse effects are reported in section 3.4, showing that the most frequent effect was pain at the injection site
  3. V4 mentioned in line 266 should be corrected to V3.

Response: This potential confusion has now been clarified, as follows: Two women did not receive the third vaccine but provided blood samples for analysis at V4.

  1. Besides mentioning the percentage of individuals having antibodies against the HPV after the vaccination as mentioned in the Immunogenicity section 3.5, the presentation of antibody titers at different sample collection time points in the form of the figure should be included. The authors have mentioned the same in the table 5. However, the depiction of such important data in the form of a figure/graph would be better. The manuscript already has a lot of tables. The inclusion of some figures especially important data would improve the quality of the presentation and hence interest to the readers.

Response: As recommended, we have now included a graph that depicts the median antibody titer values at the collection time points over the first seven months.

  1. The authors should discuss the significant changes in the clinical variables like SGPT and Bilirubin (Table 5) after the vaccination. This could indicate some adverse effects on the liver functions. The authors have not mentioned these significant changes anywhere in the manuscript. This is very necessary considering the safety of the vaccine.

SGOT, mean (±SD)

SGPT, mean (±SD)

GGT, mean (±SD)

AP, mean (±SD)

Bilirubin, mean (±SD)

Creatinine, mean (±SD)

21.24 (7.82)

20.76 (10.41)

39.56 (120.12)

77.85 (47.03)

0.71 (1.01)

0.81 (0.19)

22.28 (8.51)

20.84 (8.69)

37.34 (110.53)

79.06 (58.88)

0.61 (0.62)

0.84 (0.48)

22.39 (9.98)

22.31 (11.20)

48.13 (218.51)

79.15 (52.32)

0.56 (0.27)

0.84 (0.51)

23.63 (16.85)

23.84 (21.23)

49.62 (189.40)

83.45 (56.18)

0.57 (0.25)

1.49 (±6.81)

0.051

0.012

0.060

0.191

0.036

0.64

Response: This was not mentioned because the changed values were always within the laboratory’s range of normality.

  1. The data like CD4/CD8 ratio and viral load seem to have strong fold changes after the vaccination but no significant. This could be due to higher variation between the individuals. I would suggest presenting those data in graphical form like a scatter plot to make the reader easily understandable. The way of presentation of results can be improved by expressing them in figures whichever is possible instead of many tables. Figures reflect the data better than the tables and make it more easily understandable

Response: CD4/CD8 and viral load values remained stable, as now depicted in Figure 2. The highest viral load was recorded after the first vaccine dose, when three women were in virologic failure (see table)

Reviewer 3 Report

Comments and Suggestions for Authors

Carmen Hidalgo-Tenorio and co-authors submitted a manuscript to Vaccines dedicated to analyzing the HPV vaccine in HIV-infected women in Spain. The study included 122 HIV-infected women.

The reviewer has some comments which the authors should resolve.

1) The title refers to Spanish women, and in line 312, "They typically had 312 Spanish nationality." How was nationality determined? By place of birth, parents' nationality, or self-determination?

2) The introduction is too short and does not describe what was known in the scientific literature before the authors decided to conduct the study and their motivation.

3) Figure 1 duplicates the information presented in the text of the article and would be suitable as a graphical abstract; in its current form, it is of no value to the reader and can be deleted without losing the manuscript value.

4) Most of the information in Table 1 can be transferred to the Supplement, as it is either irrelevant or not used in the text.

5) Similar to Tables 2 and 3, it can all be moved to the Supplementary.

6) Table 6 contains much unnecessary information for the reader; it should be reduced by about ten times. All the data irrelevant to the Results and Discussion sections should be moved to the Supplementary section.

7) A Conclusions section should be added to the article.

8) It would also be appropriate to include in the Introduction the hypothesis authors proposed when designing the experiments. The authors should state whether the hypothesis has been confirmed or refuted in the Conclusions section.

The article generally does not correspond to the level of "Vaccines" regarding the work done and the level of experiments performed.

Author Response

Carmen Hidalgo-Tenorio and co-authors submitted a manuscript to Vaccines dedicated to analyzing the HPV vaccine in HIV-infected women in Spain. The study included 122 HIV-infected women.

The reviewer has some comments which the authors should resolve.

  • The title refers to Spanish women, and in line 312, "They typically had 312 Spanish nationality." How was nationality determined? By place of birth, parents' nationality, or self-determination?

Response: We are grateful to the reviewer for detecting this error. It has now been clarified in the title and text that these women were residents in Spain. Only 71.3% had been born in the country, as shown in Table 1. 

  • The introduction is too short and does not describe what was known in the scientific literature before the authors decided to conduct the study and their motivation.

Response: The Introduction has been expanded as requested.

  • Figure 1 duplicates the information presented in the text of the article and would be suitable as a graphical abstract; in its current form, it is of no value to the reader and can be deleted without losing the manuscript value.

Response: If possible, we would prefer to maintain this chart, which is habitually included in clinical trial reports and provides the reader with an instant visualization of the patient flow. Nevertheless, we would willingly delete this figure if required by the editorial team.

Most of the information in Table 1 can be transferred to the Supplement, as it is either irrelevant or not used in the text.

Response: This has been done, improving the readability of the table.

  • Similar to Tables 2 and 3, it can all be moved to the Supplementary.

Response We have moved some of the data to supplementary tables (S1 and S2). However, we prefer to maintain the now-smaller Tables 2 and 3, believing that the remaining information they exhibit is essential and facilitates the readers’ understanding of our study.

  • Table 6 contains much unnecessary information for the reader; it should be reduced by about ten times. All the data irrelevant to the Results and Discussion sections should be moved to the Supplementary section.

Response: Table 6 has now been moved to a supplementary table, preserving all data to allow access to the smallest detail of our study.

  • A Conclusions section should be added to the article.

Response: Conclusions are given in lines 392- 394: In conclusion, nHPV vaccine was safe and immunogenic in Spanish adult WLHIV. The sole risk factor for anal mucosa infection due to high-risk genotypes in this population was concomitant cervical infection by the same genotypes.

  • It would also be appropriate to include in the Introduction the hypothesis authors proposed when designing the experiments. The authors should state whether the hypothesis has been confirmed or refuted in the Conclusions section.

Response: The study hypothesis has now been added in lines 85-87: “With this background, the working hypothesis of this study was that the nonavalent HPV vaccine is immunogenetic and safe in adult WLHIV residing in Spain.“

Our confirmation of this hypothesis is reported in the Conclusions (see above).

  • The article generally does not correspond to the level of "Vaccines" regarding the work done and the level of experiments performed.

Response: We believe that our manuscript is worthy of publication in Vaccines as the first clinical trial worldwide of the Gardasil-9 vaccine in WLHIV. We have fulfilled all requirements for a robust clinical trial in relation to sample size, methodology, and statistical analysis. The vaccine is approved by the European and Spanish Drug Agencies.

Round 2

Reviewer 2 Report

Comments and Suggestions for Authors

The authors have addressed all the earlier queries and concerns.

Author Response

Comments and Suggestions for Authors: The authors have addressed all the earlier queries and concerns. Reponse: Thank you very much for your comments. 

Reviewer 3 Report

Comments and Suggestions for Authors

The authors corrected the issues in the manuscript. However, the paper still does not correspond to the level of "Vaccines" regarding the amount of work done and the level of experiments performed.

Author Response

Comments and Suggestions for Authors: The authors corrected the issues in the manuscript. However, the paper still does not correspond to the level of "Vaccines" regarding the amount of work done and the level of experiments performed.

Responses:We believe that our manuscript is worthy of publication in Vaccines as the first clinical trial worldwide of the Gardasil-9 vaccine in WLHIV. We have fulfilled all requirements for a robust clinical trial in relation to sample size, methodology, and statistical analysis. The vaccine is approved by the European and Spanish Drug Agencies.